# Quercetin Regulates Lipid Metabolism and Fat Accumulation by Regulating Inflammatory Responses and Glycometabolism Pathways: A Review

**DOI:** 10.3390/nu16081102

**Published:** 2024-04-09

**Authors:** Yaodong Wang, Zezheng Li, Jianhua He, Yurong Zhao

**Affiliations:** College of Animal Science & Technology, Hunan Agricultural University, Changsha 410128, China; sxllwyd@gmail.com (Y.W.); l15093587113@163.com (Z.L.)

**Keywords:** quercetin, obesity, lipid metabolism, inflammation, glycometabolism

## Abstract

Fat synthesis and lipolysis are natural processes in growth and have a close association with health. Fat provides energy, maintains physiological function, and so on, and thus plays a significant role in the body. However, excessive/abnormal fat accumulation leads to obesity and lipid metabolism disorder, which can have a detrimental impact on growth and even harm one’s health. Aside from genetic effects, there are a range of factors related to obesity, such as excessive nutrient intake, inflammation, glycometabolism disease, and so on. These factors could serve as potential targets for anti-obesity therapy. Quercetin is a flavonol that has received a lot of attention recently because of its role in anti-obesity. It was thought to have the ability to regulate lipid metabolism and have a positive effect on anti-obesity, but the processes are still unknown. Recent studies have shown the role of quercetin in lipid metabolism might be related to its effects on inflammatory responses and glycometabolism. The references were chosen for this review with no date restrictions applied based on the topics they addressed, and the databases PubMed and Web of Sicence was used to conduct the references research, using the following search terms: “quercetin”, “obesity”, “inflammation”, “glycometabolism”, “insulin sensitivity”, etc. This review summarizes the potential mechanisms of quercetin in alleviating lipid metabolism through anti-inflammatory and hypoglycemic signaling pathways, and describes the possible signaling pathways in the interaction of inflammation and glycometabolism, with the goal of providing references for future research and application of quercetin in the regulation of lipid metabolism.

## 1. Introduction

According to the World Health Organization (WHO), obesity has more than doubled since 1990. In 2022, 43% of adults aged 18 years and over were overweight, and 16% were living with obesity. Overweight and obesity have spread almost worldwide and are closely related to health risks [1]. Numerous studies have reported the effects of overweight and obesity on various diseases, including cardiovascular diseases [2], gastrointestinal diseases [3], metabolic diseases [4] and others. The conditions may contribute to increased disability and mortality rates [2,5]. Moreover, obesity has a detrimental impact on breeding and increases healthcare expenditures for humans [6,7]. The cost caused by overweight and obesity increases every year [8,9]. According to the investigation [10], the economic impacts of overweight and obesity are projected to increase to 3.29% of Gross Domestic Product (GDP) globally by 2060 (from 2.19% of 2019). Overweight is defined as body mass index (BMI) greater than or equal to 25 kg/m^2^; and obesity is defined as BMI greater than or equal to 30 kg/m^2^ [11]. There is evidence that obesity is associated with a complex interaction that includes inflammation, glycometabolism, and so on [12,13,14]. Inflammation can increase fat storage by blocking insulin production, lowering insulin sensitivity, and disrupting energy homeostasis. The body’s conversion of carbohydrates to fat is influenced by the synthesis and breakdown of glucose; excessive intake or synthesis of glucose enhances the conversion of carbohydrates to fat; the disorder of glycometabolism was once thought to be a major contributing factor to the disorder of lipid metabolism. Furthermore, obesity also induces inflammation and the disorder glycometabolism. Fat accumulation in adipose tissues increases the synthesis of free fatty acid (FFA), which increases the secretion of cytokines by raising and activating macrophage polarization, leading to inflammation. Excessive FFA also directly enhances the program of gluconeogenesis in the liver. Overall, there exists a complex relationship between obesity, inflammation, and glycometabolism. Alleviating inflammation and maintaining the balance of glycometabolism may play a role in anti-obesity efforts.

Polyphenols are secondary metabolites found in plants, vegetables, cereals, fruits, coffee and tea, among other sources. They are categorized based on their chemical structures into flavonoids, including flavones, flavonols, isoflavones, and more. Polyphenols are commonly used in human nutrition and animal husbandry because of their various biological activities. Many studies have demonstrated that polyphenols and their metabolites can reduce fat storage in adults [15], by modulating multiple signaling pathways related to inflammation or glycometabolism, but the mechanisms are still unknown. Quercetin, also known as 3,3′,4′,5,7-pentahydroxyflavone, is a dietary flavonoid found in vegetables, fruits, medicinal herbs, and tea [16,17,18]. Due to its distinctive chemical structure, quercetin possesses a variety of biological properties, including anti-microbial [19,20], anti-oxidant [21], anti-inflammatory [22] and blood glucose regulation [23,24] properties. It also plays a significant role in health management and has been investigated as a potential conditioner for controlling lipid metabolism. It has shown significant lipid-lowering effects in vitro [25], obese mice [26], and overweight/obese patients [27,28], which may be associated with mechanisms such as quercetin’s anti-inflammatory properties, modulation of intestinal microbiota imbalance and related gut–liver axis activation, and regulation of glucose metabolism. This review discusses how quercetin influences lipid metabolism by regulating inflammation and glycometabolism. Furthermore, the potential signaling pathways in the interplay of inflammation and glycometabolism are described.

## 2. Materials and Methods

The references chosen for this review were based on scholarly search engines, including Pubmed and Web of Science. The search keywords were meticulously selected, focusing on crucial topics, including “quercetin”, “obesity”, “inflammation”, “glycometabolism”, “insulin sensitivity”, etc. We concentrated mainly on journal articles that underwent an intense review process and excluded preprints and conference papers in this domain. The motivation was to guarantee the reliability and validity of this review. The temporal scope of references had no restrictions. This was done with the intention of identifying pertinent articles for a comprehensive review.

## 3. Quercetin: Chemical Structure, Absorption and Metabolism

Quercetin is a flavonol compound with a yellow hue that is interconnected by two aromatic rings through the carbon frame C6-C3-C6, is entirely soluble in lipids and alcohols, and mildly soluble in water. As illustrated in Figure 1, quercetin is found in nature in the form of its derivatives, most notably glycosides (rutin) [29,30]. Much research had discovered quercetin’s significant antioxidant ability, which is largely determined by its chemical structure, including catechol and hydroxyl groups [31,32]. Previous research has demonstrated the antioxidant capability of these groups by directly eliminating free radicals in vivo [33].

Numerous studies have investigated the intestinal absorption of quercetin, and the primary absorption mechanisms are as follows [34,35,36]: After ingestion, quercetin is absorbed into the epithelial cells of the small intestine via passive diffusion or facilitated by organic anion transport polypeptides. Subsequently, it undergoes phase II metabolism through a series of enzymatic reactions. A portion of quercetin that is not absorbed by the small intestine enters the large intestine and is degraded into metabolites by bacteria. Following a series of physiological processes, these compounds are taken into the liver via blood circulation and eventually eliminated via feces or urine. Furthermore, a study revealed that a proportion of quercetin free aglycones can be absorbed into the bloodstream trough the stomach, even though quercetin and its derivatives typically exhibit stability in gastric acid and are not readily absorbed [37].

It should be noted that the bioavailability of quercetin in various forms varies in the body. Bioavailability is defined as the specific amount of a substance reaching the intended site of action, and the bioavailability of quercetin plays a critical mediator in their bioactivity. Hollman et al. [38] discovered that quercetin glucosides derived from onions had a bioavailability of 52% ± 15% in the human ileum. In comparison, quercetin rutinoside showed a bioavailability of 17% ± 15%, and quercetin aglycone from tea demonstrated a bioavailability of 24% ± 9%. According to Ader et al. [39], quercetin glucuronide and quercetin sulfate were more readily absorbed in pigs’ intestinal walls. The variance in quercetin absorption may be due to factors such as quercetin derivative solubility, the type of sugar groups linked to quercetin, and the sugar coupling sites [40]. Various technical methods, including enzyme modification, nanotechnology, and target carriers, have been identified in relevant research as effective approaches for enhancing the bioavailability of quercetin [1,41,42].

## 4. Quercetin and Lipid Metabolism

### 4.1. Inflammation

The inflammatory response (inflammation) occurs when tissues are injured by bacteria, trauma, toxins, heat, or other causes. Inflammation is a typical procedure in the body to demonstrate the effect of infections that can be harmful to health and limit growth. Previous research has shown that quercetin has substantial anti-inflammatory activity in vitro and in vivo [43]. Sun et al. [44] and Gruse et al. [45] found that quercetin down-regulated the expression of inflammatory factors, and it suggested the potential of quercetin to increase livestock and poultry anti-inflammatory capacity. In another study, quercetin was discovered to inhibit the production of inflammatory mediators induced by lipopolysaccharide (LPS) in the macrophage [46,47,48]. Quercetin can also reduce the synthesis of inflammatory factors by inhibiting the phosphorylation of related inflammatory enzymes and enhancing the activity of antioxidant enzymes [49,50]. 

Inflammation is caused by the interaction of several factors, one of which is the inflammation caused by obesity. As a potential anti-inflammatory agent, quercetin can be utilized to treat inflammation caused by fat accumulation and obesity. A study demonstrated that the inflammatory factor in adipocytes, the expression level of interleukin-1 (IL-1), and interleukin-6 (IL-6), was dramatically down-regulated after utilizing quercetin in vitro tests and obese mice models. Further investigation found that quercetin may have prevented the activation of the signaling pathways for the proteins mitogen-activated protein kinase (MAPK), extracellular regulated protein kinases (ERK), and C-Jun N-terminal kinase (JNK), which are involved in the production of inflammatory mediators in adipocytes [25]. In addition, the results also indicated that the weight of obese mice was significantly decreased after feeding with quercetin, and the expression of proteins of the key adipose factors including CCAAT/enhancer binding protein (C/EBP), peroxisome proliferators-activated receptor γ (PPARγ) and fatty acid-binding protein 4 (FABP4) as well as synthetases of triglyceride (TG) were significantly down-regulated [25]. Yang et al. [51] also found that quercetin can effectively inhibit the production of IL-1β, IL-6 and tumor necrosis factor α (TNF-α) induced by type 2 diabetes mellitus (T2DM). Overall, quercetin may decrease fat accumulation by regulating various inflammation signaling pathways. The following summarizes the potential mechanisms of quercetin in regulating inflammation, based on previous studies (Figure 2).

Quercetin can activate the PPARγ signaling pathway as well as inhibit the activity of inflammatory factors induced by MAPK signaling pathway, which activates leptin signaling in adipose tissue and accelerates fat oxidation [52]. Furthermore, PPARγ is a crucial early regulator of fat production that is strongly tied to adipocyte differentiation; it can also trigger adiponectin expression, which can regulate the formation of mature adipocytes [53,54]. As a result, quercetin can either negate the effects of antagonism caused by inflammatory stimuli on PPARγ or directly activate PPARγ to increase its anti-obesity efficacy.

Besides, due to its effect on MAPK signaling pathways, quercetin can reduce the production of FFA via farnesoid X receptor 1 (FXR1)/Takeda G protein-coupled receptor 5 (TGR5) and ERK/JNK signaling pathways [55]. FXR/TGR5 can mediate bile acid signaling pathways, which is beneficial for the oxidation of Fatty Acid (FA) [56,57]. Phosphorylation of ERK/JNK inhibits the apoptosis of mature adipocytes and enhances the deposition of fat [58]. In addition, previous studies have indicated that the effects of quercetin on inflammation and lipid metabolism may also be regulated by insulin, and its mechanism may be related to factors such as glycometabolism, Insulin Resistance (IR) and so on [59,60,61]. The following sections summarize the potential mechanisms of quercetin in regulating glycolipid metabolism.

Furthermore, our findings indicate that quercetin might decrease the levels of inflammatory factors through the inhibition of mast cell activation. This mechanism may contribute to the anti-inflammatory properties of quercetin. Li et al. [62] conducted a study where they induced THP-1 cell differentiation, macrophage migration, and P815 cell activation, followed by treatment with quercetin. They found that quercetin significantly inhibited the differentiation of THP-1 cells into macrophages and the macrophages polarization. P815 cell activation was inhibited when treated with quercetin. Dong et al. [60] found that quercetin decreased mast cell infiltration in HFD mice. Zhao et al. [63] also reported that quercetin can inhibit the activation and recruitment of mast cells in HFD mice. These studies indicate that quercetin has the potential to impede macrophage infiltration and mast cell activation. The existing literature predominantly examines the impact of quercetin on mast cells and macrophages in vitro or in mice. However, the current literature lacks adequate evidence to support the effects of quercetin on mast cells in obese patients or those with abnormal glycometabolism. Therefore, further research is needed.

### 4.2. Glycometabolism

Glucose is the main source of energy in the body and its homeostasis is closely related to health. It is known that there is a close relationship between glycometabolism and fat accumulation. In normal conditions, glucose is stored in the liver as glycogen, while excessive glucose is converted into fat. Excessive intake of glucose may break the balance of glycometabolism and decrease insulin sensitivity, which may cause to lipid metabolism disorders and increased fat accumulation. Numerous studies have reported the potential effects of quercetin and its derivatives on hypoglycemic [64]. Manzano et al. [65] demonstrated that quercetin-3-O-rhamnoside derived from strawberries and apples could inhibit glucose absorption in intestinal Caco-2 cells, primarily by inhibiting Sodium-dependent glucose transporters 1 (SGLT1) and recombinant glucose transporter 2 (GLUT2). Pico et al. [66] also mentioned that quercetin decreased the glucose absorption activity of the intestinal brush border membrane by inhibiting glucose transporters SGLT1 and GLUT2. In addition to its direct impact on glucose absorption, quercetin can activate a number of signaling pathways to modulate insulin production and IR, thereby lowering blood glucose levels. In T2DM mice, Li et al. [67] discovered that quercetin could efficiently boost insulin levels. Subsequent research revealed that quercetin may also prevent iron-induced pancreatic islets β Cell death and restore pancreatic islet cell dysfunction. Tan et al. [68] fed HFD male mice with quercetin, and the results indicated that weight, glucose concentration and the homeostasis model assessment of insulin resistance (HOMA-IR) were significantly down-regulated, possibly due to the inhibition of the glucose transduction signal pathway GLUT4. The study also found that the expression of FA-metabolism-related genes stearoyl-CoA desaturase 1 (SCD1) and sterol regulatory element binding transcription factor 1 (Srebf1) was dramatically down-regulated. SCD1 is one of the important targets of carbohydrate response element-binding protein (ChREBP), which can regulate whole-body lipid metabolism by controlling the transcription of lipogenic enzymes and liver-derived cytokines, and it mediates glucose’s induction of glycolysis [69,70]. Related studies [70,71] have shown that a deficiency of SCD1 increases FA oxidation and insulin sensitivity. Srebf1 is a kind of transcription factor that regulates lipid homeostasis by controlling the expression of a range of enzymes, and it can encode transcription factors that increase glycolysis and lipogenesis [69,72]. Therefore, the effect of quercetin in glycometabolism might be mediated by GLUT4, SCD1 and Srebf1 signaling pathways.

In addition to its direct effect on glycometabolism, quercetin can indirectly regulate glycometabolism through anti-inflammatory pathways, owing to its effect on IR [73]. Inflammation in tissues is a major contribute to IR, and there is evidence that TNF-α has a beneficial influence on fat storage and IR [74]. As described in the previous section, quercetin could alleviate the release of TNF-α induced by fat accumulation, hence inhibiting IR. Leptin and adiponectin are critical regulators of adipocyte differentiation. Adiponectin is an important target of PPAR-γ, and binding of adiponectin to its receptor can improve insulin sensitivity [75]. According to a recent study, leptin could reduce glucose levels and prevent the synthesis of FA [76]. Yadav et al. [77] also observed that leptin increased FA oxidation while decreasing glucose levels, which was conveyed via the janus kinase/signal transducer and activator of transcription (JAK/STAT) signaling pathway. Furthermore, quercetin promotes liver gluconeogenesis by inhibiting the MAPK signal pathway, which causes the dephosphorylation of forkhead box transcription factor O1 (FOXO1) [78,79] and inhibits leptin signal transduction [80]. Figure 3 summarizes the probable mechanism by which quercetin regulates glycolipid metabolism.

The synthesis and degradation of glucose are deemed to play a significant role in the metabolism of FA [81]. Glucose is the main source of energy for the cells. Excessive glucose is converted into fat by the procedures of de novo synthesis of FA, resulting in the release of large amount of FA into the blood and tissues. In healthy conditions, high blood sugar increases the release of insulin and is suppressed; however, as blood sugar levels rise, cells’ sensitivity to insulin declines, ultimately resulting in IR. High blood glucose levels promote the synthesis of fat, while excessive fat accumulation promotes the synthesis of glucose even more [82,83,84]. According to Vidal et al. [85], HFD significantly worsened whole-body glucose tolerance, increased total body fat mass, and compromised metabolic performance in mice. Mizuno et al. [86] discovered a consistent connection between the expression of genes associated with obesity and glucose production. According to Astrup et al. [87], hyperglycemia may hinder the regulating impact of a reduced carbohydrate diet on fat synthesis and breakdown. Therefore, the hypoglycemic effect of quercetin may also be one of the effective ways for quercetin to exert its anti-obesity activity.

## 5. Inflammation, Glycometabolism and the Related Signaling Pathways

As previously stated, obesity-induced FFA upregulation promotes the release of inflammatory factors, which leads to macrophage invasion of adipose cells and stimulation of M1 polarization, thereby activating other pro-inflammatory cytokines or inhibiting the release of protective adipocytokines. Inflammation can induce pancreatic islets β Cell apoptosis, disrupt insulin signaling pathway transduction, and impair normal mitochondrial function, all of which eventually lead to IR. Meanwhile, hyperglycemia-induced IR can activate the activities of inflammatory factors, leading to inflammation in cells and tissues. Overall, there is a complicated connection between inflammation and glycometabolism. However, how is the interaction transmitted? Are there any shared genes or signaling pathways? Recent studies have found the role of many signaling pathways in the connection between inflammation and glycometabolism. The following sections mainly discuss the potential mechanisms of FOXO1 in inflammation and glycometabolism.

### 5.1. FOXO1

FOXO1 is a transcription factor that belongs to the FOXO family. It plays crucial roles in cell cycle control, apoptosis, metabolism and adipocyte differentiation. It is known that FOXO1 affects adipocyte differentiation by regulating lipogenesis and the cell cycle. In addition, previous research has shown that FOXO1 participates in the development of both inflammation and IR, and that it may co-regulate inflammation and glucose signaling pathways in vivo. FOXO1 was discovered to be involved in the regulation of macrophage activation and polarization [88]. FOXO1 inhibited macrophage M2 polarization by antagonizing the Signal transducer and activator of transcription 6 (Stat6). Stat6 targets PPAR in macrophages, which helps activate M2 polarization in macrophages [89]. Meanwhile, FOXO1 promotes the expression of IL-1β and TLR4 in macrophages, enhancing M1 polarization [90,91]. Interestingly, FOXO1 plays a crucial role in inflammatory signaling pathways in adipose cells and tissues by modulating macrophage M1 and M2 polarization. Previous research has discovered that insulin can reduce FOXO1 activity. The upstream signaling pathway of FOXO1 is protein kinase B (Akt), which mediates insulin signaling transport through insulin receptor substrates (IRSs). In normal physiological conditions, insulin and IL4 can act on IRSs and suppress FOXO1 activity via the Akt-dependent pathway [92]. While the inhibition of insulin on FOXO1 is lost in IR states, this ultimately leads to inflammation. 

Furthermore, FOXO1 also plays a role in initiating gluconeogenesis processes. The activation of MAPK decreases the phosphorylation of FOXO1 and stimulates FOXO1 translocation from the cytosol to the nucleus [78]. After that, activated FOXO1 interacts with peroxisome proliferator-activated receptor-gamma coactivator 1 (PGC-1α). PGC-1α is an inducible coactivator for nuclear hormone receptors and other transcriptional factors. It is a key gene in gluconeogenesis procedures [93]. Gu et al. [94] also demonstrated that the FOXO1/PGC-1 pathway is involved in regulating the expression and activation of gluconeogenic enzymes in both HFD-induced rats and IR cells. Puigserver et al. [95] found that the regulation of PGC-1α in gluconeogenic procedures is primarily dependent on the interaction of PGC-1α and FOXO1. This effect is inhibited by insulin treatment in normal states. Insulin treatment can mediate FOXO1 phosphorylation by activated Akt, leading to its sequestration in the cytoplasm through the insulin signal transduction pathway. This ultimately disrupts the FOXO1-PGC-1α interaction.

Moreover, FOXO1 has been found to be related to pancreatic β-cells dysfunction which impairs insulin secretion and consequently leads to glycometabolism disorder. Wang et al. [96] found that long-term glucagon intervention induced β-cell dedifferentiation, and that the effect was partially mediated by FOXO1. Similar to the study conducted by Talchai et al. [97] in β-cells of mice with IR diabetes, the loss of FOXO1 with increased hyperglycemia induced β-cells differentiation. However, it should be noted that the loss of FOXO1 does not alter β-cells death or self-renewal in vivo. This may indicate that β-cells dysfunction is not related to the potential proapoptotic role of FOXO1. In conclusion, FOXO1 plays an important regulatory role in various pro-inflammatory and glucose signaling pathways, suggesting that FOXO1 is a valuable therapeutic target for the treatment of obesity.

### 5.2. Quercetin and FOXO1

FOXO1 plays a role in the regulation of inflammation, glycometabolism and insulin signal transduction. It is a component of a synergistic signaling pathway that connects these three processes. As previously discussed, FOXO1 plays a role in the regulation of inflammation, IR and gluconeogenesis. It plays an important role in regulating inflammation induced by obesity or abnormal glycometabolism. As mentioned earlier, the mechanism of quercetin in decreasing fat deposition is related to its effect of anti-inflammatory and hypoglycemic effects. Quercetin can inhibit FOXO1 dephosphorylation, it is induced by inflammatory signaling pathways, such as MAPK. This indicates that FOXO1 is one of the important signaling pathways through which quercetin mediates inflammation and glucose signaling pathways. It is also one of the key targets of quercetin’s anti-obesity ability. Other studies have also reported the effect of quercetin on FOXO1. For example, He et al. [98] carried out in vivo and in vitro experiments using quercetin. The results indicated that quercetin can activate FOXO1 in pulmonary arterial smooth muscle cells (PASMCs) under hypoxic conditions, inducing PASMCs apoptosis. Liu et al. [99] reported that quercetin can alleviate IR and decrease glucose production by inhibiting FOXO1 expression. The study also revealed that the inhibition of quercetin on gluconeogenesis was significantly inhibited after knocked-out FOXO1. It suggests that FOXO1 may be a key target to quercetin’s effect on glucose signal transduction. The relevant studies have reported that the mediation of FOXO1 by quercetin appears to mainly rely on its upstream genes, such as PI3K, IRS and Akt [99,100]. By inhibiting inflammation, quercetin can modulate the activity of these related genes, thereby influencing FOXO1 regulation. It should be noted that there is currently no evidence to confirm whether quercetin can regulate the transcriptional activity and the expression levels of FOXO1 through other signaling pathways. Further research is needed.

## 6. Conclusions

Obesity has a significant impact on human and animal health due to its close relationship with metabolic illnesses. Obesity is closely related to inflammation and dysregulation of glycometabolism. Inflammation and abnormal glycometabolism contribute to lipid metabolic issues and increased fat formation. As a result, inflammation and glycometabolism may be potential targets for reforming aberrant lipid metabolism and reducing fat formation. Quercetin is a type of polyphenol found in nature that is known to have antioxidant capabilities due to its hydroxyl groups. It is frequently used in anti-aging products and diets. Furthermore, quercetin is recognized as a promising product in health management due to its anti-inflammatory, hypoglycemic and other effects. Anti-obesity compounds can influence differentiation, synthesis and lipolysis by modulating signaling pathways such as inflammation signaling pathways (NF-κb, MAPK, AP-1, etc.), glycometabolism signaling pathways (PGC-1α, ChREBP, GLUT1, etc.), adipocytokines (leptin, adiponectin), and so on.

Signaling pathways, such as FOXO1, have been found to be involved in both inflammation development and glycometabolism. They can regulate macrophage activation and polarization through Stat6 and ATF3, which play a role in the development of inflammation in adipose tissues. Furthermore, they can also play a role in regulating glycometabolism by modulating PGC-1α and IRS. As a result, FOXO1 could be a promising target for anti-obesity treatment. Although FOXO1 may be a crucial target in the treatment of metabolic illnesses, such as T2DM, hyperglycemia, etc., there is little evidence to support the effect of quercetin on FOXO1. The relationship between quercetin and the important signaling pathway FOXO1 needs to be explored further. In conclusion, quercetin is a potent molecule in regulating inflammation and glycometabolism. It has anti-obesity effects and may play a role in the treatment of metabolic illnesses.

## Figures and Tables

**Figure 1 nutrients-16-01102-f001:**
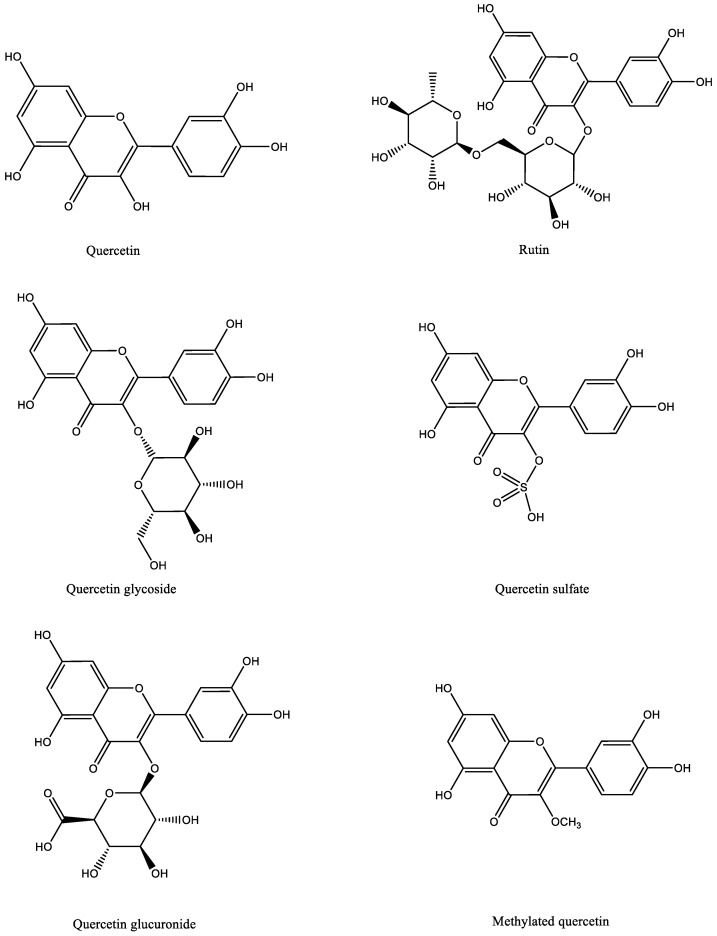
Molecular structure of quercetin, rutin, and its common derivatives.

**Figure 2 nutrients-16-01102-f002:**
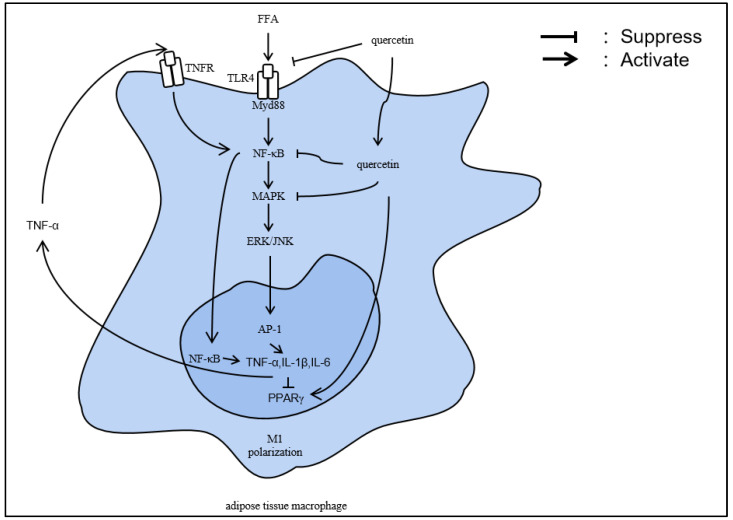
Potential mechanisms of quercetin regulating inflammation. FFA, free fat acid; TLR4, Toll-like receptor 4; MyD88, myeloiddifferentiationfactor88; NF-κb, nuclear transcription factor-kappa B; MAPK, mitogen-activated protein kinase; ERK, extracellular regulated protein kinases; JNK, c-Jun n-terminal kinase; TNFR, tumor necrosis factor receptor; TNF-α, tumor necrosis factor α; IL-1β, interleukin-1β; IL-6, interleukin-6; AP-1, activator protein 1; and PPARγ, Peroxisome Proliferators-activated Receptor γ; MI, classically activated macrophage.

**Figure 3 nutrients-16-01102-f003:**
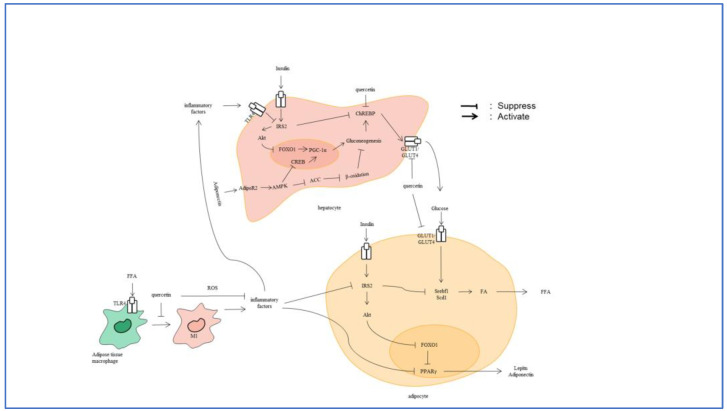
The possible mechanism of quercetin regulating glycolipid metabolism. FA, fat acid; FFA, free fat acid; ROS, reactive oxygen species; IRS2, insulin receptor substrate 2; Akt, protein kinase b; FOXO1, forkhead box transcription factor O1; Srebf1, sterol regulatory element binding transcription factor 1; Scd1, stearoyl-CoA desaturase 1; PPARγ, Peroxisome Proliferators-activated Receptor γ; GLUT1, recombinant glucose transporter 1; GLUT4, recombinant glucose transporter 4; ChREBP, carbohydrate response element-binding protein; TLR4, Toll-like receptor 4; AdipoR2, adiponectin receptor 2; AMPK, AMP-activated protein kinase; ACC, acetyl CoA carboxylase; PGC-1α, peroxisome proliferative activated receptor-gamma co-activator 1; and CREB, cAMP-response element binding protein; MI, classically activated macrophage.

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
