# Peer review of "Quercetin Regulates Lipid Metabolism and Fat Accumulation by Regulating Inflammatory Responses and Glycometabolism Pathways: A Review"

_nutrients, 2024, doi:10.3390/nu16081102_

Round 1

Reviewer 1 Report

Comments and Suggestions for Authors

The review manuscript authored by Yaodong Wang et al. provides an insightful overview of the potential role of the flavanol Quercetin in regulating lipid metabolism through its anti-inflammatory and hypoglycemic effects. However, there are some areas that require clarification and improvement.

Specific points

1.      Most individuals that are obese or diabetic had decreased insulin receptor surface expression.   Is the effect of Quercetin on insulin signaling pathway independent of insulin receptor expression level?

2.      The link between Quercetin and Baf60a is highly speculative and lacks support from previous published evidence.

3.      Are there any published effect of Quercetin on mast cell number in obese/T2 diabetes mellitus patients.

Minor point

Line 81: Posterior intestine – authors probably refer to large intestine since the bacterial degradation is mentioned. Careful usage of anatomical terminology, particularly when referencing parts of the intestine, is advised.

Comments on the Quality of English Language

Among the minor shortcomings, I can note a small number of typos (for example in line 301); unnecessary word capitalization and the redundant use of phrases like »and so on«.

Reviewer 2 Report

Comments and Suggestions for Authors

Dear authors,

In general the manuscript is well whiten.

Here are my remarks:

1. Introduction: it must be revised. Make it more profound and detailed.

The first sentence must be modified: "Obesity is characterized by abnormal or excessive fat buildup that might harm one's health." This sentence does not have scientific soundness.

Include more data about obesity and overweight. Strategies for prevention or management.

You could check these manuscripts:

Popkin, B.M.; Du, S.; Green, W.D.; Beck, M.A.; Algaith, T.; Herbst, C.H.; Alsukait, R.F.; Alluhidan, M.; Alazemi, N.; Shekar, M. Individuals with obesity and COVID-19: A global perspective on the epidemiology and biological relationships. Obes. Rev. 202021, e13128. 

Ivanova, S.; Delattre, C.; Karcheva-Bahchevanska, D.; Benbasat, N.; Nalbantova, V.; Ivanov, K. Plant-Based Diet as a Strategy for Weight Control. Foods 202110, 3052. https://doi.org/10.3390/foods10123052

Grasemann, H.; Holguin, F. Oxidative stress and obesity-related asthma. Paediatr. Respir. Rev. 202037, 18–21. 

You could check the website of the World Health Organisation:

https://www.who.int/news-room/fact-sheets/detail/obesity-and-overweight

2. The manuscript needs an English editing. It is better to be performed by a native English speaker.

3. Could you explain better the search procedure for inclusion the articles in your review? For example, is it possible to include a PRISMA figure?

http://prisma-statement.org/?AspxAutoDetectCookieSupport=1

4. Results/Discussion:

Could you include a table which summarises your findings about studies about the relationship between quercetin and obesity/overweight management?

5. The conclusions must be modified:

  1.  
  • This sentence is not correct and it is misleading "Obesity is known to be caused by inflammation and a dysregulation of glycometabolism."
  • In general obesity provokes inflammation.
  • Obesity could be a result of many factors: such as misbalance in calorie intake and energy expenditure
Comments on the Quality of English Language

A moderate editing of English language is required.

Round 2

Reviewer 1 Report

Comments and Suggestions for Authors

The authors significantly improved the manuscript and addressed the raised points adequately. 

Author Response

Thank you for your letter and comments concerning our manuscript entitled “Quercetin regulates lipid metabolism and fat accumulation by regulating inflammatory responses and glycometabolism pathways: A Review” . Those comments are all valuable and very helpful for revising and improving our paper, as well as the important guiding significance to our researches. We sincerely thank you for taking the time to review our manuscript.

Reviewer 2 Report

Comments and Suggestions for Authors
  1. Dear authors,

  2. here are my comments:
  3. 1. Table 1. "The relationship between quercetin and obesity/overweight management" is too short. In my view you could include the data from the table in the introduction and you can remove the table. However, if you decide to save the table, the references should be the last column, not the first one.

  4. 2. The search procedure should be included not only in the abstract but also in the main text. You could include Materials and Methods section and to describe it in this section.
  5.  
  6. 3. Format the references according to MDPI guidelines (check size, font, etc).
